# Morphological, molecular, and pathological characterization of didymozoid trematode infection in the nasal cavity of orange-spotted grouper (*Epinephelus coioides*) from Arabian Gulf waters

**Mohsen A. Khormi[1], Mohamed Abdelsalam[2], Hanadi B. Baghdadi[3], Mustafa M. Ibrahim[4], Mahmoud A. Mahmoud[5], Marwa M. Attia[6]***

1 Department of Biology, College of Science, Jazan University, Jazan, Kingdom of Saudi Arabia,
2 Department of Aquatic Animal Medicine and Management, Faculty of Veterinary Medicine, Cairo University, Giza, Egypt, 3 Department of Biology, Faculty of Science, Imam Abdul Rahman bin Faisal University, Dammam, Saudi Arabia, 4 Deptartment of pathology, Animal Health Research Institute, Giza, Egypt, 5 Department of Pathology, Faculty of Veterinary Medicine, Cairo University, Giza, Egypt, 6 Department of Parasitology, Faculty of Veterinary Medicine, Cairo University, Giza, Egypt

* marwaattia.vetpara@yahoo.com, marwaattia.vetpara@cu.edu.eg

## Abstract

### Background

Didymozoid trematodes represent poorly understood parasites of marine fish worldwide, with significant knowledge gaps in the Arabian Gulf region. The taxonomic identification of didymozoids from novel anatomical sites requires integrated morphological and molecular approaches to resolve species boundaries and phylogenetic relationships.

### Methods

We examined 2,530 fish specimens from nine *Epinephelus* species collected from Arabian Gulf waters (Saudi Arabia). Parasitological, histopathological, and ultrastructural analyses were combined with molecular characterization using 28S rDNA sequences. Phylogenetic relationships were inferred using Maximum Likelihood analysis with 1,000 bootstrap replicates.

### Results

52 of 500 *Epinephelus coioides* (10.4%) harbored distinctive golden-yellow cysts located solely in nasal cavities. No infections were found in other grouper species or anatomical sites. Histopathology revealed multiple cysts containing the didymozoid parasites, with characteristic numerous eggs; and the cysts were surrounded by thin layer of fibrous connective tissue capsule with mild host inflammatory response. Ultrastructural examination demonstrated characteristic didymozoid features

**Data availability statement:** All relevant data are within the paper and its Supporting Information files.

**Funding:** The author(s) received no specific funding for this work.

**Competing interests:** The authors have declared that no competing interests exist.

including bean-shaped eggs and specialized tegument structures. Molecular analysis of four seasonal isolates yielded two haplotypes (99.81% similarity) with 1,042 nucleotide sequences deposited as GenBank accessions PQ736510-PQ736513. Phylogenetic analysis confirmed placement within *Didymodiclinus* (Didymozoidae), showing 99.52% sequence homology with *Didymodiclininae* sp. from the same host species.

## Conclusions

This study provides the first morphological, molecular, and histopathological characterization of nasal cavity didymozoids in *E. coioides* from Saudi Arabian waters of the Arabian Gulf. The infections were restricted to *E. coioides* among nine grouper species examined and occurred only in nasal cavities, demonstrating remarkable host and anatomical site specificity. These findings provide baseline data for *didymozoid* ecology in Arabian Gulf waters.

## Introduction

Adult digenetic trematodes of the family Didymozoidae (Poche, 1907) represent one of the most specialized groups of parasitic flatworms, with over 270 described species parasitizing more than 100 marine fish families in tropical and subtropical waters worldwide [1,2]. Unlike most digenetic trematodes that inhabit body cavities or organ lumens, didymozoids have evolved unique adaptations for histozoic parasitism, forming characteristic encapsulated cysts within host tissues including gills, musculature, body cavity, subcutaneous tissues, and internal organs [3,4]. This ecological specialization has resulted in extraordinary morphological diversity, characterized by extreme sexual dimorphism and specialized structures for nutrient acquisition within encapsulated microenvironments [5,6].

The unusual morphology and encapsulated lifestyle of didymozoids present significant taxonomic challenges that have historically complicated species identification and phylogenetic reconstruction [7,8]. Traditional morphological approaches often prove insufficient due to the compressed, distorted nature of specimens within host tissue capsules, requiring complementary molecular approaches for reliable species delimitation [9,10]. The large subunit ribosomal RNA gene (28S rDNA) has emerged as a valuable molecular marker for didymozoid systematics, with the D1-D3 expansion segments providing sufficient resolution for species-level identification and phylogenetic inference [11,12]. Recent studies using integrative taxonomic approaches that combine morphological, molecular, and ecological data have identified previously unrecognized species within didymozoid lineages, indicating that current species richness estimates may substantially underestimate true diversity [13,14].

Despite the rich marine biodiversity of the Arabian Gulf, comprehensive surveys of didymozoid fauna remain limited, with only scattered reports documenting species occurrence and distribution patterns [15,16]. The Arabian Gulf's unique environmental conditions, including elevated salinity, temperature extremes, and geographic

isolation from other marine regions, may promote endemic parasite evolution and novel host-parasite associations [17,18]. Previous investigations by Abdul-Salam et al. [19,20] described didymozoid species from *Epinephelus tauvina* in Kuwait waters, but systematic surveys across the broader Gulf region remain incomplete. This knowledge gap is particularly significant given the economic importance of grouper fisheries and the potential impact of parasitic infections on fish health, marketability, and aquaculture sustainability [21,22].

The orange spotted grouper (*Epinephelus coioides*, Hamilton 1822) represents one of the most commercially valuable fish species in Arabian Gulf fisheries, supporting both traditional capture fisheries and emerging aquaculture operations [23,24]. This predatory species has a relatively long lifespan and occupies diverse habitats from shallow reefs to deeper offshore waters, serving as an important reservoir for various parasite species and providing insights into marine ecosystem health [22,24]. The broad Indo-Pacific distribution and complex life cycle of *E. coioides* make it a suitable model for understanding geographic patterns of parasite diversity and host-parasite evolutionary relationships [25,26].

Members of the family Didymozoidae are highly specialized histozoic digenean trematodes that induce distinctive pathological alterations in their fish hosts as a consequence of their encapsulated lifestyle within host tissues [3,4]. Unlike luminal trematodes, didymozoids form paired or fused cysts embedded in organs such as gills, musculature, body cavity, and sensory structures, where chronic mechanical pressure, local tissue displacement, and nutrient competition may occur [5,12]. Pathogenesis is typically characterized by cyst formation surrounded by fibrous connective tissue capsules, reflecting a host containment response rather than acute inflammation [2]. Although inflammatory reactions are often mild due to parasite localization in poorly vascularized tissues, heavy infections have been associated with tissue distortion, impaired organ function, reduced respiratory efficiency (in gill infections), and compromised sensory or feeding performance depending on infection site [3,4,8]. Subclinical impacts, including reduced growth, altered behavior, and increased susceptibility to secondary infections, have been suggested in heavily parasitized hosts [14]. Consequently, didymozoid infections may exert cumulative negative effects on host fitness, particularly in commercially important fish species, highlighting their ecological and economic relevance despite often inconspicuous clinical presentation [12,25].

Modern parasitological studies increasingly rely on integrative approaches that combine traditional morphological taxonomy with molecular phylogenetics, advanced imaging techniques, and quantitative pathological assessment [27,28]. Molecular phylogenetic analysis using multiple gene markers provides robust frameworks for species identification and evolutionary inference, while high-resolution microscopy techniques reveal structural details crucial for taxonomic characterization [11,12]. The integration of seasonal sampling strategies with molecular analysis enables investigation of temporal genetic variation and population structure within parasite populations. This multidisciplinary approach proves particularly valuable for didymozoid studies, where extreme morphological specialization and encapsulated lifestyles complicate traditional taxonomic approaches [29,30].

This study aimed to provide comprehensive characterization of didymozoid parasites infecting *E. coioides* in Arabian Gulf waters through integrated morphological, pathological, molecular, and ultrastructural analysis. Specific objectives included: (1) documenting infection prevalence and intensity across seasonal samples; (2) characterizing gross and histopathological changes associated with parasitic infection; (3) describing ultrastructural features using transmission electron microscopy; (4) conducting molecular identification using 28S rDNA sequencing and phylogenetic analysis; and (5) evaluating taxonomic relationships with previously described *Didymozoid* species.

## Materials and methods

### Ethics statement

The collection and analysis adhered to guidelines from the Veterinary Office within the Welfare of Fisheries division in Jubail Province, Saudi Arabia.

## Fish sample collection and examination

During January to December 2024, routine fish health surveillance was conducted at Jubail Fish Market, Eastern Province, Saudi Arabia (27°02'20.5"N, 49°38'18.9"E). Local fishermen collected specimens from Arabian Gulf waters using traditional fishing methods. A total of 2,530 fish specimens representing nine *Epinephelus* species were examined for parasites, morphological anomalies, and pathological conditions. Among these, 500 specimens of *E. coioides* (Hamilton, 1822) were specifically targeted for detailed parasitological examination. Fish specimens showing distinctive golden-yellow cysts in nasal cavities were immediately transported to the laboratory in insulated containers with ice for comprehensive analysis.

## Parasitological analysis

Parasite cysts were carefully extracted from nasal cavities of infected fish using sterile forceps and dissecting needles under a stereomicroscope. Cyst dimensions were measured using digital calipers, and morphological characteristics were documented through digital photography. For taxonomic examination, didymozoid cysts were dissected and fixed in 5% neutral buffered formol saline to preserve the delicate cyst capsule structure without over-hardening for 24 hours, then transferred to 70% ethyl alcohol for long-term storage following standard protocols [5]. Taxonomic procedures followed established guidelines for didymozoid processing [31].

## Histopathological examination

For histopathological evaluation, the cysts from nasal region were gently detached with the tissue remnants of nasal mucosa at its attachment sites (to ensure inclusion of the cysts together with the adjacent nasal mucosa). The existed cysts with the tissue of the nasal mucosa were immediately fixed in 10% neutral buffered formalin, processed routinely through graded ethanol for dehydration, cleared in xylene, and embedded in paraffin wax. Serial sections (4–5 μm) were cut on a rotary microtome and stained with hematoxylin and eosin (H&E) for general histopathological assessment. Slides were examined under light microscopy at multiple magnifications to document cyst architecture, capsule characteristics, host tissue response, and associated lesions [32].

## Ultrastructural Analysis

For transmission electron microscopy (TEM), fresh specimens were rinsed in physiological saline and immediately fixed in 2.5% glutaraldehyde in 0.1M sodium cacodylate buffer (pH 7.2) at 4°C overnight. Samples were post-fixed in 1% osmium tetroxide at 4°C for 24 hours, then dehydrated through a graded ethanol series (50–100%) and embedded in epoxy resin. Semi-thin sections (500–1000 nm) were cut using a Leica ultracut (UCT ultramicrotome) and stained with toluidine blue for examination at Cairo University College of Agriculture. Ultrathin sections (70–90 nm) were cut, mounted on copper grids, and stained with uranyl acetate and lead citrate for TEM examination. Transmission electron microscopy was performed using a JEOL JEM-1400 TEM operated at 80 kV. Digital images were captured at various magnifications with appropriate scale bars and calibration standards.

## Molecular identification

Molecular analyses were performed on four individual didymozoid specimens extracted from four different infected host fish (*E. coioides*) collected across different seasons. Each parasite was processed individually (not pooled) to detect genetic variation.

## DNA extraction

Genomic DNA was extracted independently from whole parasite specimens using the Qiagen DNeasy Tissue Kit (Qiagen, Hilden, Germany) following manufacturer protocols with modifications for small tissue samples. Specimens were

incubated with proteinase K overnight at 56°C in a rotating incubator to ensure complete tissue digestion. DNA was eluted in 200 μL elution buffer and concentrated to approximately 20 μL using Millipore Microcon columns (Merck Millipore, Massachusetts, USA) when necessary.

## PCR Amplification of 28S rDNA

The partial D1-D3 fragment of 28S rDNA was amplified using primers LSU5 (5′-TAGGTCGACCCGCTGAAYTTAAGC-3′ [11] and 1500R (5′-GCTATCCTGAGGGAAACTTCG-3′) [10]). PCR reactions were performed in duplicate in 20 μL volumes containing: 5 μL of 5×MyTaq Reaction Buffer (Bioline, Australia), 0.75 μL of each primer (10 μM), 0.25 μL MyTaq DNA Polymerase (Bioline), 4 μL template DNA (~20 ng), and ultrapure distilled water to final volume. PCR conditions included initial denaturation at 95°C for 4 minutes, followed by 40 cycles of denaturation (95°C, 30 s), annealing (50°C, 30 s), and extension (68°C, 2 min), with final extension at 68°C for 7 minutes [33]. The expected PCR product size was approximately 1,000–1,100 bp, and amplification success was verified by 1% agarose gel electrophoresis.

## Sequencing and phylogenetic analysis

PCR products were visualized on 1% agarose gels and purified using QIAquick Gel Extraction Kit (QIAGEN). Purified amplicons were bidirectionally sequenced using BigDye Terminator v3.1 Cycle Sequencing Kit (Applied Biosystems) on an ABI PRISM 3130 automated sequencer. Raw sequences were edited and assembled using BioEdit v7.2.5 [34], with chromatograms manually inspected for quality verification. Sequence consistency between technical replicates was confirmed prior to downstream analyses. Sequences were compared against GenBank using BLASTN searches. Multiple sequence alignment was performed using Clustal W, and phylogenetic relationships were inferred using Maximum Likelihood analysis in MEGA11 [35]. The General Time Reversible model with gamma-distributed rates and invariant sites (GTR+G+I) was selected based on lowest Bayesian Information Criterion and Akaike Information Criterion scores. Bootstrap support was assessed using 1,000 replicates.

## Statistical analysis

Infection prevalence was calculated as the percentage of infected fish among total examined specimens. Seasonal variation in infection rates was compared using Chi-square tests. Statistical significance was set at $P < 0.05$. All analyses were performed using basic statistical functions.

## Inclusivity in global research

An inclusivity questionnaire is provided as Supporting Information, addressing the ethical, cultural, and scientific aspects relevant to ensuring inclusivity within this global research.

## Results

### Infection prevalence and seasonal distribution

Among 2,530 examined fish specimens representing nine *Epinephelus* species, only *E. coioides* harbored didymozoid parasites. Of 500 examined *E. coioides*, 52 individuals (10.4%) showed characteristic golden-yellow cysts exclusively in nasal cavities (Table 1). No infections were observed in other anatomical sites or fish species. Seasonal infection rates showed moderate variation: summer 11.2% (14/125), autumn 10.7% (16/150), winter 10.0% (13/130), and spring 9.5% (9/95), with no statistically significant seasonal differences ($P > 0.05$); Supplementary file. Infected fish typically harbored 3.0±0.8 cysts per nostril, with cyst diameters averaging 2.4±0.4 mm (Table 1).

**Table 1. Seasonal prevalence and characteristics of Didymozoid spp. infections in *Epinephelus coioides* from Arabian Gulf waters, Saudi Arabia.**

| Season | Fish Examined | Infected | Prevalence (%) * | Cysts per Nostril** | Cyst Diameter (mm)** |
|---|---|---|---|---|---|
| Summer | 125 | 14 | 11.2 | 3.1±0.8 | 2.4±0.3 |
| Autumn | 150 | 16 | 10.7 | 2.9±0.7 | 2.5±0.4 |
| Winter | 130 | 13 | 10.0 | 3.2±0.9 | 2.3±0.3 |
| Spring | 95 | 9 | 9.5 | 2.7±0.6 | 2.6±0.5 |
| Total | 500 | 52 | 10.4 | 3.0±0.8 | 2.4±0.4 |

*No significant seasonal variation in infection prevalence.

**Mean±SD in infected fish.

**Note:** Of 2,530 fish examined from nine *Epinephelus* species, only *E. coioides* showed didymozoid infections. All cysts were restricted to nasal cavities. While the other eight species were parasite-free: *Cephalopholis rogaa* (n = 300), *C. miniatus* (n = 220), *E. areolatus* (n = 250), *E. bleekeri* (n = 320), *E. multinotatus* (n = 250), *E. octofasciatus* (n = 340), *E. polylepis* (n = 200), and *E. caeruleopunctatus* (n = 250).

## Gross pathological findings

The gross pathological examination of the affected Grouper in the fish markets (Fig 1A) showed the presence of didymozoid parasitic cysts within the nostril cavity. The parasitic cysts appeared as distinctive golden-yellow, oval to hemispherical structures measuring 2–3 mm in diameter and were located exclusively within the nasal cavities of infected fish (Fig 1B&C). The cysts exhibited smooth, fleshy walls with bright yellow coloration and contained thread-like worms measuring several centimeters in length. During gross examination, the cysts were readily visible and could be easily distinguished from normal nasal tissue (Fig 1D). No external gross signs of inflammation or tissue damage were observed around cyst attachment sites. Impression smears of the cysts revealed numerous characteristic ovoid eggs typical of didymozoid parasites (Fig 1E).

## Histopathological findings

The histopathological examination revealed that the didymozoid cyst contained multiple encysted worms in 2–4 fused cysts (Fig 2A). The cysts were aggregated as separate entities, each encapsulated with a thin layer of fibrous connective tissue and containing prominent parasitic structures. The large, thick female worm contained numerous eggs, while the thin male displayed testicular tissue. The fibrous capsule was thickened at the site of attachment to the host nasal mucosa (Fig 2B&C). The eggs within the female worm, located inside the ovaries, appeared at different stages of maturation (Fig 2D&E), while others appeared degenerated without internal structures. Some examined cysts were small with delicate fibrous connective tissue capsules, whereas others showed thickening of the capsule with more degenerated eggs, indicating older cyst formation (Fig 2F).

## Ultrastructural observations

**Host tissue.** Transmission electron microscopy revealed a complex host-parasite interface characterized by an undulating epithelial layer forming the outer boundary in contact with nasal cavity tissues (Fig 3). The host tissue contained various cytoplasmic organelles including small mitochondria, large kidney-shaped nuclei, and extensive rough endoplasmic reticulum with associated fibroblasts and lysosomes. Elongated, multinucleated structural cells were observed throughout the interface region (Fig 3).

**Parasite body.** The parasite body exhibited a thin tegument (approximately 5 μm) consisting of an external basal lamina in contact with a non-cellular cyst wall layer (Fig 3). This layer contained a dense matrix of secretory granules

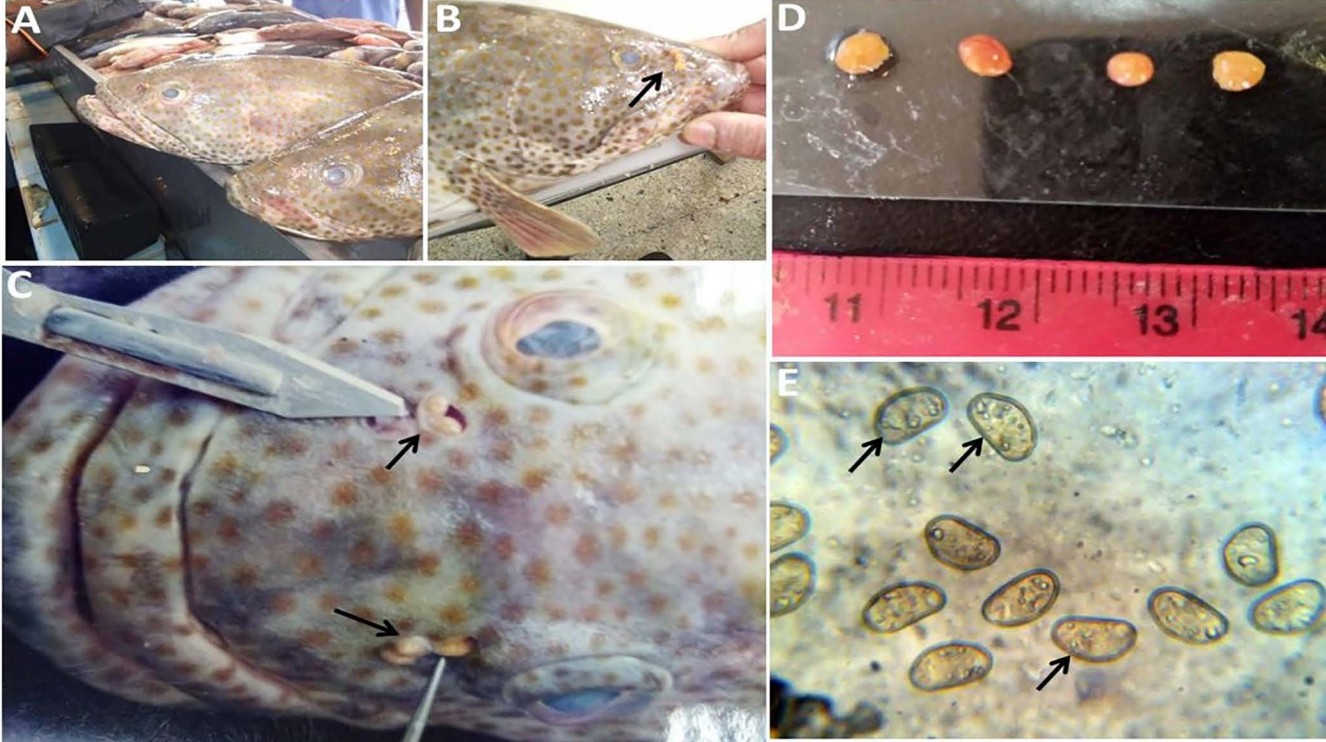

**Fig 1. Gross pathological examination of didymozoid cysts in orange spotted grouper, *Epinephelus coioides* from Arabian Gulf waters. (A)** Examined grouper fish specimens in the fish markets showing normal external appearance. **(B)** Beaded-shape (3-4) large cysts of the didymozoid unilateral in the nostril cavity (arrow). **(C)** Bilateral cyst formation on the nostrils of orange spotted grouper (arrows). **(D)** The cysts appeared as golden yellow round small structures, about 2-3 mm in diameter. **(E)** Impression smear of the cyst demonstrates the presence of many characteristic eggs of the female didymozoid parasite.

alternating with small vesicles and probable microtubule filaments. A second, thicker basal lamina separated this structure from underlying multicellular tissue layers. Bean-shaped eggs filled uterine lumens, with eggs at various developmental stages. Advanced-stage eggs exhibited increased cytoplasmic organelles (Fig 4), while earlier stages contained undifferentiated cellular masses within amorphous matrices surrounded by tight shells (Fig 4). No opercular structures were visible at the observed developmental stages.

**Molecular characterization.** Molecular analysis of four seasonal isolates yielded high-quality 28S rDNA amplicons. Sequencing produced four complete sequences of 1,042 nucleotides each, deposited in GenBank under accession numbers PQ736510-PQ736513. Sequence analysis revealed two distinct haplotypes: haplotype 1 (PQ736510, PQ736511) and haplotype 2 (PQ736512, PQ736513), differing by only two nucleotide positions (99.81% sequence identity).

BLAST analysis positioned sequences within the family Didymozoidae, specifically within the genus *Didymodiclinus* (Pozdnyakov, 1993). The closest match was *Didymodiclininae* sp. (GenBank accession OL335996.1) from *E. coioides* branchial infections, showing 99.52% sequence homology, suggesting these represent closely related lineages that may differ in anatomical site preference. Additional significant matches included 98.04% identity with *Didymozoid* sp. (AY222194.1) and 97.89% with *Didymodiclinus* sp. (OL335999).

**Phylogenetic analysis.** Maximum Likelihood analysis revealed that newly sequenced *Didymodiclinus* sp. specimens (PQ736510, PQ736512) formed a strongly supported monophyletic clade (bootstrap value 99%) most closely related

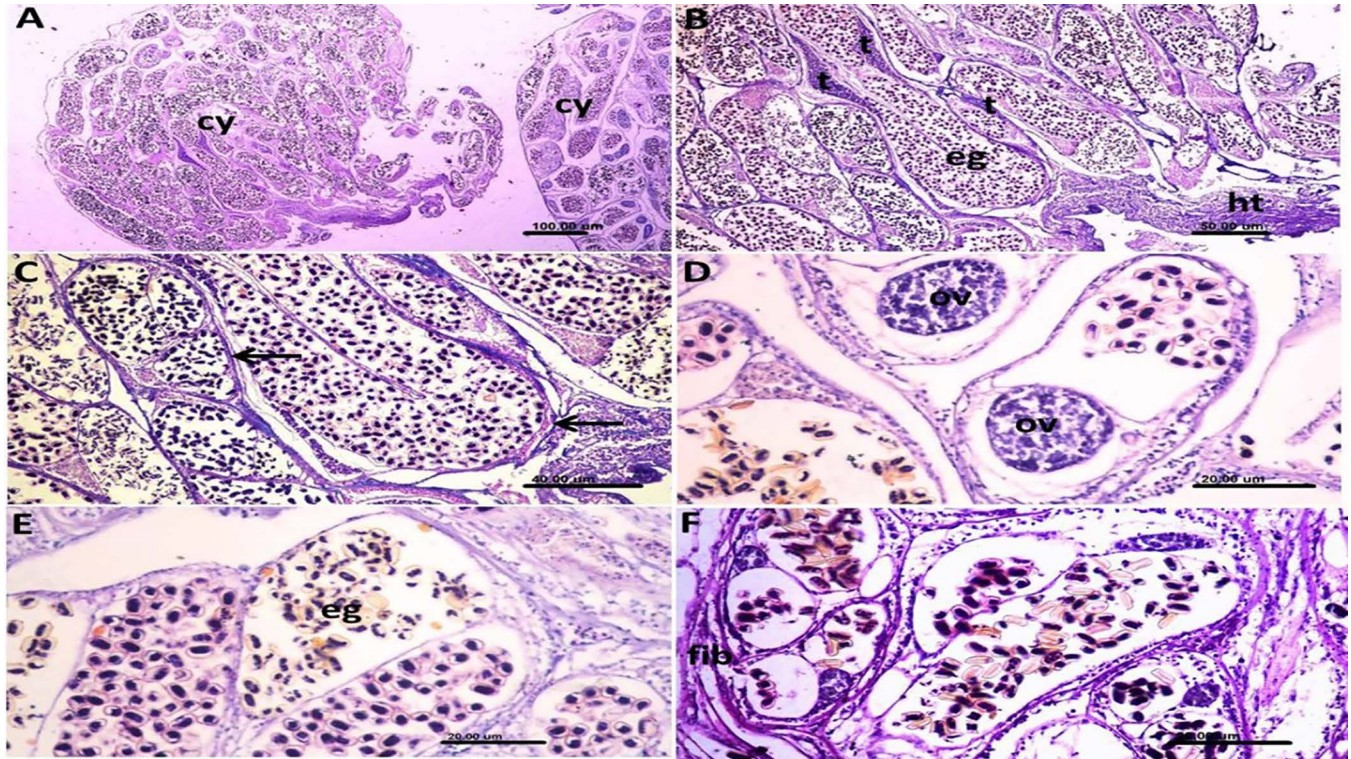

**Fig 2. Photomicrograph of the histopathological sections of *didymozoid* cysts from the nostrils of orange-spotted grouper, *Epinephelus coioides*. A:** Two parasite cysts (cy) containing multiple encysted worms separated by thin layer of fibrous connective tissue capsule. **B:** The cysts were separated by thin contained prominent many eggs (e.g.,) of the female parasite and testicular tissue (t) of the males; notice, thickening of the proliferated fibrous tissue at the site of cyst attachment to the host tissue (ht). **C:** Higher magnification of the previous photo showing the fibrous connective tissue capsule (arrows) separating the encysted worms. **D:** The eggs inside the parasite appeared with different stages of maturation near the ovarian tissue (ov). **E:** Some of the eggs (e.g.,) contained larvae and others appeared degenerated without internal structures. **F:** The fibrous capsule (fib) was thickened at some parts of the cysts indicating old cyst formation.

to *Didymodiclininae* sp. (OL335996) (Fig 5). This clade was sister to a group containing *Didymodiclinus pacificus* (OL335998) and other Didymozoidae species (OL336057, MW489507, OL335997, AY222193) with high statistical support (bootstrap value 99%). The phylogenetic tree topology demonstrated clear separation from other major didymozoid lineages, including *Nematobothrium scombri* (AY222195) and *Didymocystis scomberomori* (KU341979).

## Discussion

In this study, the restricted occurrence of didymozoid infections to *E. coioides* among nine examined grouper species demonstrates remarkable host specificity characteristic of this parasitic family [34,35]. This host-specific pattern aligns with previous observations of didymozoid-grouper associations, where Abdul-Salam et al. [19,20] reported similar specificity in *E. tauvina* from Kuwait waters. However, the current findings represent the first comprehensive molecular and histopathological characterization of didymozoids from *E. coioides* in Saudi Arabian waters. While Abdul-Salam et al. [19,20] described *Gonapodasmius epinepheli* from *E. tauvina* in Kuwait based on morphological features, our study differs in geographic location (Saudi Arabia), host species (*E. coioides*), infection site (nasal cavity vs. body cavity/gills), and methodological approach (integrating molecular phylogenetics with morphological diagnosis). This represents the first record of nasal cavity-specific didymozoid infections in groupers from the Arabian Gulf region. The restriction to nasal cavity sites represents an unusual anatomical preference, as most didymozoid species typically encyst in gill filaments, musculature,

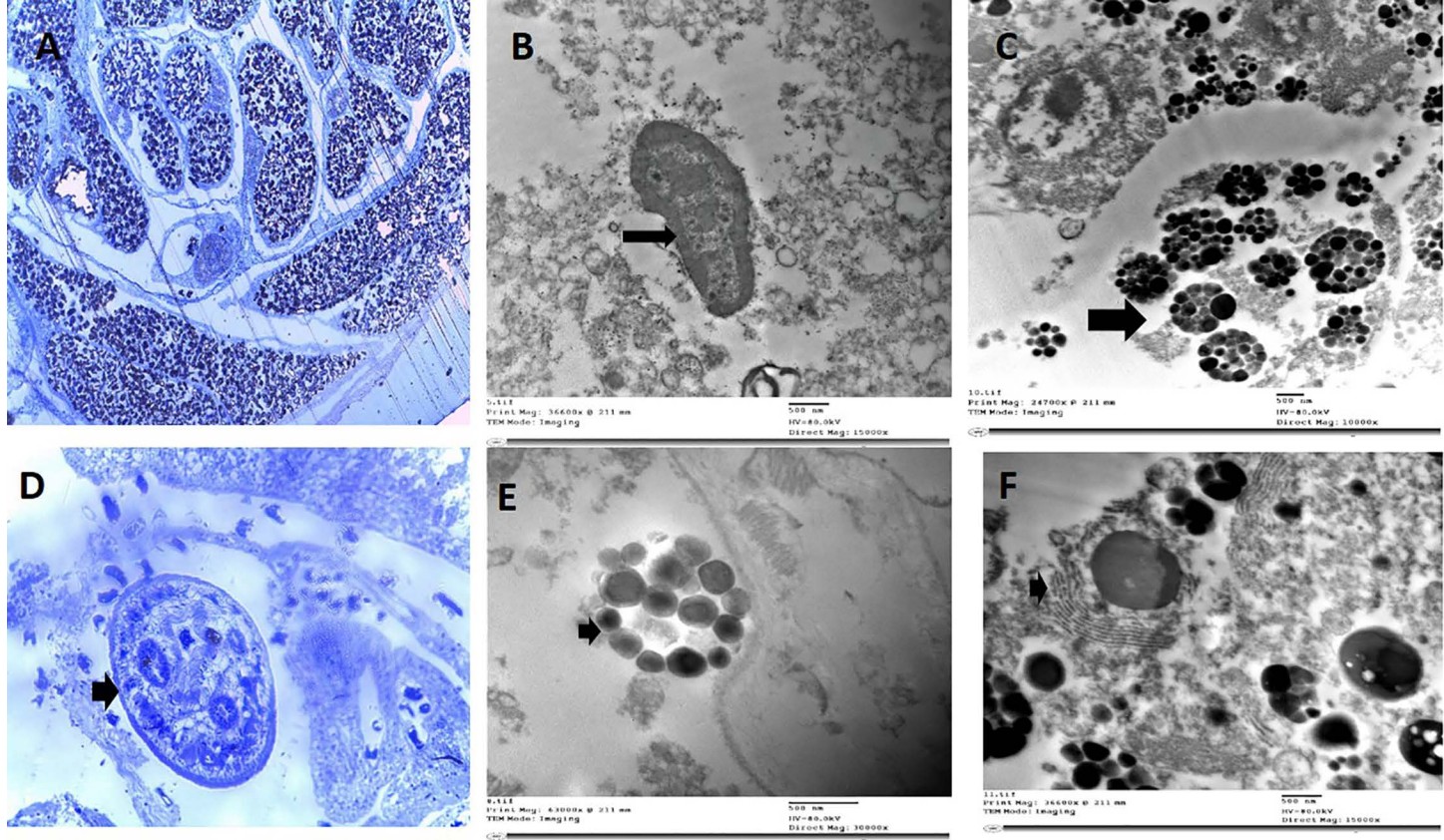

**Fig 3. *Didymodiclinus* sp. semithin and ultrathin sections of the whole cyst. A:** Semithin section showing the thick cyst wall that is in contact with the cyst's external side, which is probably touching the nasal mucosa. **B:** Ultrathin section showing small mitochondria and large kidney-shaped nuclei. **C:** Ultrathin section showing fibroblasts scattered throughout the cyst. **D:** Semithin section showing the male separated from female. **E:** Ultrathin section showing large lysosomes within a thin rim of cytoplasm. **F:** Ultrathin section showing extensive rough endoplasmic reticulum.

or body cavity tissues [3,4]. This unique site specificity may reflect specialized adaptations for nutrient acquisition and environmental conditions within the nasal cavity microenvironment. The consistent infection pattern across seasonal samples (10.4% overall prevalence) suggests stable host-parasite dynamics rather than episodic transmission events. The absence of significant seasonal variation (P > 0.05) contrasts with many marine parasite systems where environmental factors drive temporal fluctuations in infection patterns [36,37]. This stability may indicate continuous parasite transmission throughout the year, possibly related to the consistent environmental conditions of Arabian Gulf waters and the complex life cycle requirements of didymozoid species [38]. Molecular analysis revealed two distinct haplotypes (99.81% similarity) across seasonal samples, indicating low but detectable intraspecific genetic diversity within the local parasite population.

This study represents the first comprehensive molecular characterization of didymozoid parasites from Saudi Arabian waters, making a significant contribution to our understanding of Arabian Gulf marine parasite biodiversity. The molecular characterization using 28S rDNA provides robust phylogenetic placement within the subfamily Didymodiclininae, with 99.52% sequence similarity to *Didymodiclininae* sp. from the same host species. This high genetic similarity raises important questions about cryptic species diversity within didymozoid lineages, as morphologically similar parasites may represent distinct evolutionary lineages [39]. The discovery of two distinct haplotypes (99.81% similarity) from seasonal samples indicates potential population structure or early-stage speciation processes within local parasite populations [8], revealing substantial genetic diversity that likely reflects historical population changes and ongoing gene flow patterns across the region. The

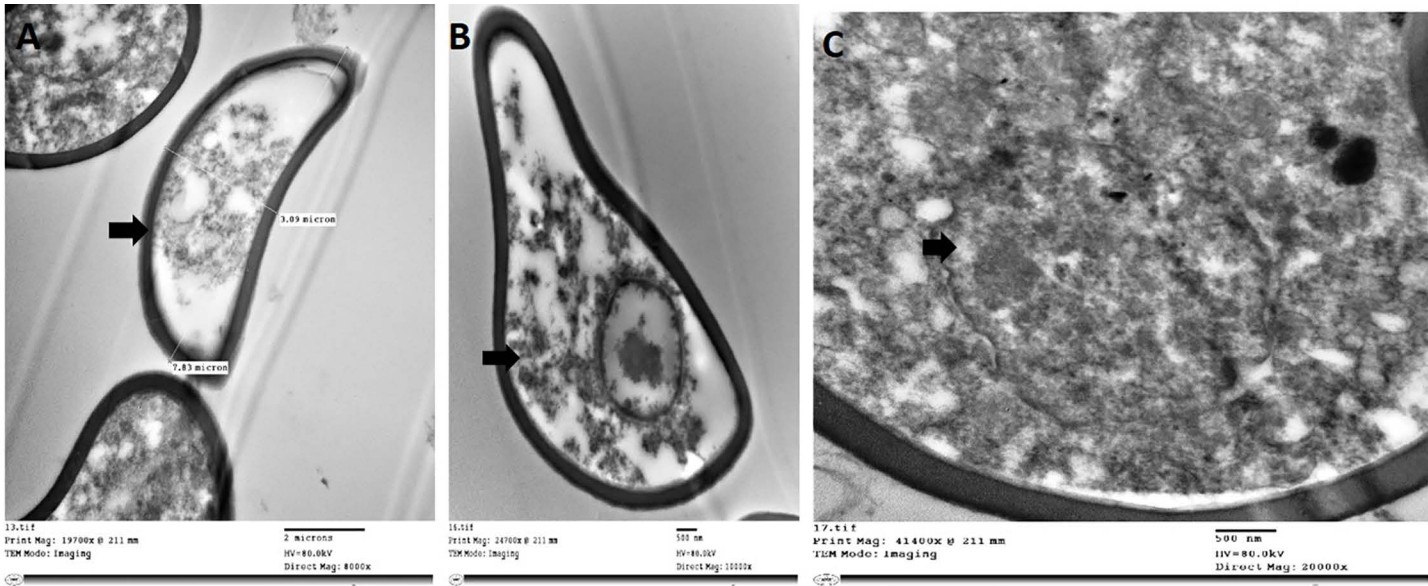

**Fig 4. Ultrathin structure of didymozoid reproductive elements. (A, B)** Ultrathin structure showing bean-shaped eggs filling the uterine lumen; some eggs in more advanced stages exhibit additional cytoplasmic organelles, while others have tight shells containing undifferentiated cells within amorphous mass. No operculum was visible at this developmental stage. **(C)** Higher magnification showing large nucleus in mature egg structure.

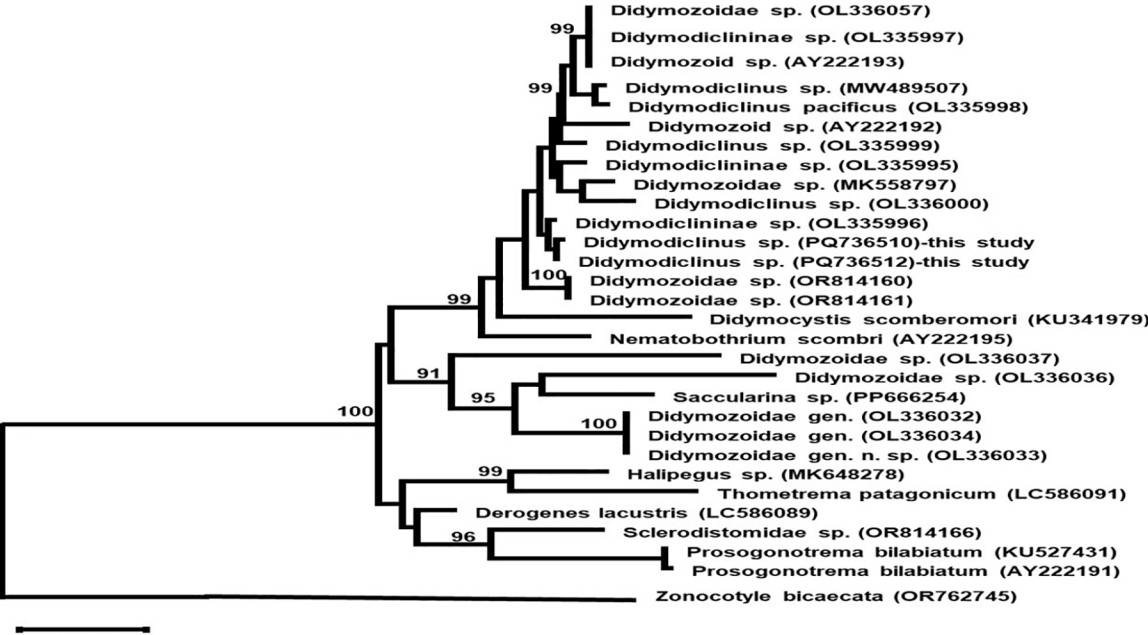

**Fig 5. Maximum likelihood phylogenetic tree of didymozoid trematodes based on partial 28S rDNA sequences.** The tree includes newly sequenced *Didymodiclinus* sp. representing two haplotypes (PQ736510 representing haplotype 1, and PQ736512 representing haplotype 2) with other members of the Didymodiclininae subfamily. Bootstrap values >90% from 1,000 replicates are shown at nodes. *Zonocotyle bicaecata* was used as the outgroup. Scale bar indicates 0.02 substitutions per site.

phylogenetic analysis confirms the monophyletic nature of nasal cavity didymozoids, supporting their distinction from gill-parasitizing relatives within the same genus and demonstrating that anatomical site specificity serves as a significant evolutionary driver in didymozoid diversification [7–11]. The restriction of infections to *E. coioides* among closely related grouper species further supports potential coevolutionary processes specific to Arabian Gulf populations, where unique environmental conditions including elevated temperatures and salinity levels drive endemic parasite evolution and novel host-parasite associations [17,18]. The molecular evidence consistently supports the biogeographic distinctiveness of Arabian Gulf didymozoid populations, with sufficient genetic divergence from Indo-Pacific relatives to confirm regional evolutionary processes shaped by the Gulf's distinctive marine environment. While 28S rDNA alone has known limitations for resolving some closely related species [7,8], it has proven effective for genus-level identification and phylogenetic placement within Didymozoidae [11,12,39]. Our analysis confidently places the nasal cavity parasites within *Didymodiclinus* (99.52% similarity to congeneric species), consistent with morphological diagnosis. Future multi-marker studies incorporating ITS2 and COI would provide additional resolution for fine-scale population structure and cryptic diversity assessment [27,28].

The minimal pathological response observed in infected fish indicates a well-established host-parasite relationship characteristic of chronic infections [17,18,40]. The thin fibrous connective tissue capsules surrounding parasite cysts represent typical host encapsulation responses designed to isolate foreign organisms while minimizing tissue damage [40]. The histopathological findings align with previous observations of didymozoid infections in poorly vascularized tissues, where limited blood supply restricts immune cell recruitment and inflammatory responses [29,40]. Mladineo [3] described similar minimal host reactions in didymozoids infecting avascular gill raker tissues, suggesting that site selection may represent an immune evasion strategy. While gross examination revealed no external signs of inflammation or severe tissue damage around cyst attachment sites, the relationship between these didymozoid trematodes and orange-spotted groupers represents a fundamentally parasitic association rather than commensalism. Didymozoids are obligate parasites that derive nutrition from host tissues [3–5], occupy anatomical space within the nasal cavity that may affect normal respiratory and olfactory functions, and elicit host encapsulation responses [40]. The absence of overt gross pathology should not be interpreted as indicating a benign relationship, as subclinical effects on host physiology and potential impacts on behavior or sensory function cannot be excluded [14,40]. Histopathological examination revealed microscopic tissue alterations including fibrous capsule formation and cellular responses at the host-parasite interface [29,40], indicating active interactions despite minimal acute inflammation.

The presence of both male and female worms within individual cysts confirms the hermaphroditic mating strategy typical of *Didymozoid* species [6]. The various egg developmental stages observed suggest continuous reproductive activity within established cysts, indicating successful long-term parasitic establishment [41]. The ultrastructural analysis provides novel insights into didymozoid-host interactions at the cellular level, revealing previously undocumented features of the host-parasite interface [42]. The bean-shaped egg morphology and specialized tegument structures contribute to morphological databases essential for species identification and phylogenetic reconstruction [43,44].

This study has several important limitations. The complete parasite life cycle of current didymozoid remains unknown, as intermediate hosts and larval stages were not identified. Sampling was limited to one geographic location, preventing broader conclusions about distribution patterns. Controlled infection experiments were not conducted. The molecular analysis used only one gene marker, which may not capture full genetic diversity. Future studies should focus on identifying intermediate hosts, expanding sampling across different locations, and using multiple genetic markers for more complete analysis. Long-term monitoring combining genetic identification with health assessments could help document how these host-parasite relationships change over time.

## Conclusion

This study provides the first comprehensive characterization of nasal cavity didymozoids in *E. coioides* from Saudi Arabian Gulf waters through combined morphological, molecular, and pathological analysis. The infections showed

remarkable host specificity, occurring only in *E. coioides* among nine examined grouper species, and represent the first documentation of didymozoid parasites in nasal cavities. Molecular analysis confirmed placement within *Didymodiclinus* (subfamily Didymodiclininae) with 99.52% sequence similarity to related species, while two identified haplotypes indicate genetic diversity within local populations. Histopathological examination revealed encapsulated cysts containing multiple worms with variable developmental stages. This integrative approach demonstrates the effectiveness of combining morphological and molecular methods for identifying parasite diversity in economically important marine species.

### Study limitations

This study has several limitations that should be acknowledged. First, sampling was restricted to a single geographic location, which may limit broader inference regarding regional distribution patterns within Arabian Gulf waters. Second, molecular characterization was based on a single genetic marker (28S rDNA), and inclusion of additional loci (e.g., ITS2, COI) could provide greater resolution for population structure and species delimitation. Finally, the complete parasite life cycle remains unknown, as intermediate hosts and larval stages were not investigated, precluding conclusions regarding transmission dynamics. Future studies incorporating host condition assessments (body condition factor, growth parameters), multi-locus molecular analyses, wider geographic sampling, and life-cycle investigations are recommended to strengthen understanding of didymozoid ecology and host–parasite interactions.

### Supporting information

**S1 File. Raw data (Supplementary file 1) contain the raw data of seasonal incidence.**
(DOCX)

**S2 File. Inclusivity-in-global-research-questionnaire.**
(DOCX)

**S3 File. Striking Image.**
(PPTX)

### Author contributions

**Conceptualization:** Marwa Attia.

**Data curation:** Marwa Attia.

**Formal analysis:** Marwa Attia.

**Funding acquisition:** Hanadi B. Baghdadi, Marwa Attia.

**Investigation:** Mohamed Abdelsalam.

**Methodology:** Mohamed Abdelsalam, Hanadi B. Baghdadi, Mahmoud A. Mahmoud.

**Resources:** Hanadi B. Baghdadi.

**Software:** Mohsen A. Khormi, Mohamed Abdelsalam, Mustafa M. Ibrahim.

**Supervision:** Mustafa M. Ibrahim.

**Validation:** Mohsen A. Khormi, Mustafa M. Ibrahim.

**Visualization:** Mustafa M. Ibrahim, Mahmoud A. Mahmoud.

**Writing – original draft:** Mustafa M. Ibrahim, Mahmoud A. Mahmoud.

**Writing – review & editing:** Mahmoud A. Mahmoud.

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
