## [Decision Letter · Decision Letter 0]

19 Jan 2026

PONE-D-25-51753Pathological, morphological, and molecular analysis of didymozoid trematodes infections in the nasal cavity of orange-spotted grouper (Epinephelus coioides) caught from Arabian Gulf watersPLOS One

Dear Dr. Attia,

Thank you for submitting your manuscript to PLOS ONE. After careful consideration, we feel that it has merit but does not fully meet PLOS ONE’s publication criteria as it currently stands. Therefore, we invite you to submit a revised version of the manuscript that addresses the points raised during the review process.

1. This manuscript not technically sound, and the data cannot support the conclusions. PLOS ONE is designed to communicate primary scientific research, and welcome submissions in any applied discipline that will contribute to the base of scientific knowledge. But this manuscript not adhere to the criteria for scientific research article that results show not sufficient to support the conclusion.

2. The revised manuscript needs to address each of the comments of the reviewers.

If applicable, we recommend that you deposit your laboratory protocols in protocols.io to enhance the reproducibility of your results. Protocols.io assigns your protocol its own identifier (DOI) so that it can be cited independently in the future. For instructions see: https://journals.plos.org/plosone/s/submission-guidelines#loc-laboratory-protocols. Additionally, PLOS ONE offers an option for publishing peer-reviewed Lab Protocol articles, which describe protocols hosted on protocols.io. Read more information on sharing protocols at . Additionally, PLOS ONE offers an option for publishing peer-reviewed Lab Protocol articles, which describe protocols hosted on protocols.io. Read more information on sharing protocols at https://plos.org/protocols?utm_medium=editorial-email&utm_source=authorletters&utm_campaign=protocols..

We look forward to receiving your revised manuscript.

Kind regards,

Tzong-Yueh Chen, Ph.D.

Academic Editor

PLOS One

**Journal Requirements:**

1. When submitting your revision, we need you to address these additional requirements. Please ensure that your manuscript meets PLOS ONE's style requirements, including those for file naming. The PLOS ONE style templates can be found at https://journals.plos.org/plosone/s/file?id=wjVg/PLOSOne_formatting_sample_main_body.pdf and https://journals.plos.org/plosone/s/file?id=ba62/PLOSOne_formatting_sample_title_authors_affiliations.pdf 2. Please include a complete copy of PLOS’ questionnaire on inclusivity in global research in your revised manuscript. Our policy for research in this area aims to improve transparency in the reporting of research performed outside of researchers’ own country or community. The policy applies to researchers who have travelled to a different country to conduct research, research with Indigenous populations or their lands, and research on cultural artefacts. The questionnaire can also be requested at the journal’s discretion for any other submissions, even if these conditions are not met.  Please find more information on the policy and a link to download a blank copy of the questionnaire here: https://journals.plos.org/plosone/s/best-practices-in-research-reporting. Please upload a completed version of your questionnaire as Supporting Information when you resubmit your manuscript. 3. We note that your Data Availability Statement is currently as follows: All relevant data are within the manuscript and its Supporting Information files. Please confirm at this time whether or not your submission contains all raw data required to replicate the results of your study. Authors must share the “minimal data set” for their submission. PLOS defines the minimal data set to consist of the data required to replicate all study findings reported in the article, as well as related metadata and methods (https://journals.plos.org/plosone/s/data-availability#loc-minimal-data-set-definition). For example, authors should submit the following data: - The values behind the means, standard deviations and other measures reported;- The values used to build graphs;- The points extracted from images for analysis. Authors do not need to submit their entire data set if only a portion of the data was used in the reported study. If your submission does not contain these data, please either upload them as Supporting Information files or deposit them to a stable, public repository and provide us with the relevant URLs, DOIs, or accession numbers. For a list of recommended repositories, please see https://journals.plos.org/plosone/s/recommended-repositories. If there are ethical or legal restrictions on sharing a de-identified data set, please explain them in detail (e.g., data contain potentially sensitive information, data are owned by a third-party organization, etc.) and who has imposed them (e.g., an ethics committee). Please also provide contact information for a data access committee, ethics committee, or other institutional body to which data requests may be sent. If data are owned by a third party, please indicate how others may request data access. 4. When completing the data availability statement of the submission form, you indicated that you will make your data available on acceptance. We strongly recommend all authors decide on a data sharing plan before acceptance, as the process can be lengthy and hold up publication timelines. Please note that, though access restrictions are acceptable now, your entire data will need to be made freely accessible if your manuscript is accepted for publication. This policy applies to all data except where public deposition would breach compliance with the protocol approved by your research ethics board. If you are unable to adhere to our open data policy, please kindly revise your statement to explain your reasoning and we will seek the editor's input on an exemption. Please be assured that, once you have provided your new statement, the assessment of your exemption will not hold up the peer review process. 5. Please amend either the title on the online submission form (via Edit Submission) or the title in the manuscript so that they are identical. 6. Please include your full ethics statement in the ‘Methods’ section of your manuscript file. In your statement, please include the full name of the IRB or ethics committee who approved or waived your study, as well as whether or not you obtained informed written or verbal consent. If consent was waived for your study, please include this information in your statement as well. 7. Please upload a new copy of Figures 3 and 4, as the detail is not clear. Please follow the link for more information:  https://journals.plos.org/plosone/s/figures 8. If the reviewer comments include a recommendation to cite specific previously published works, please review and evaluate these publications to determine whether they are relevant and should be cited. There is no requirement to cite these works unless the editor has indicated otherwise.

Reviewers' comments:

Reviewer's Responses to Questions

**Comments to the Author**

1. Is the manuscript technically sound, and do the data support the conclusions?

Reviewer #1: Partly

Reviewer #2: Partly

2. Has the statistical analysis been performed appropriately and rigorously? 

Reviewer #1: Yes

Reviewer #2: Yes

3. Have the authors made all data underlying the findings in their manuscript fully available?

Reviewer #1: Yes

Reviewer #2: Yes

4. Is the manuscript presented in an intelligible fashion and written in standard English?

Reviewer #1: Yes

Reviewer #2: Yes

5. Review Comments to the Author

**Reviewer #1:** Nasal cavity didymozoid trematodes were found exclusively in Epinephelus coioides from the Arabian Gulf, with integrated morphological and molecular analyses confirming their identity as Didymodiclinus and highlighting high host and site specificity. Nasal cavity didymozoid trematodes were found exclusively in Epinephelus coioides from the Arabian Gulf, with integrated morphological and molecular analyses confirming their identity as Didymodiclinus and highlighting high host and site specificity.

Here are comments to the authors

1. Line 128 and 133: Why were the didymozoid cysts fixed in 5% neutral buffered formol saline instead of 10% neutral buffered formalin, as used for the infected tissues?

2. Line165: Please specify the expected and/or obtained PCR product size.

3. Line 196-197: Since no external signs of inflammation or tissue damage were observed around the cyst attachment sites, does the relationship between the didymozoid trematodes and the orange-spotted groupers represent a parasitic, mutualistic, or commensal association?

4. Line 2: Since no external signs of inflammation or tissue damage were observed, the authors may need to reconsider the use of the terms “pathological” and “infection” in the title.

5. Is there any data available on the physiological parameters (weights and lengths) of orange-spotted groupers with and without didymozoid cysts? Such information could help determine whether the presence of didymozoid cysts affects the growth and health of orange-spotted groupers.

6. Line 58-66: Please provide more information about the family Didymozoidae and didymozoid trematodes, including aspects such as their pathogenesis, impact on the host, and associated mortality in host.

**Reviewer #2:** In general, this is a well-organized and very interesting manuscript. The authors have analyzed three categories of parasitology, i.e. pathological, morphological and molecular level investigations, for Didymozoidae in orange-spotted grouper hosts, and generated a comprehensive report. However, there are few questions need to be answered. If the authors can adequately answer these questions, I will recommend accept the manuscript for publication. In general, this is a well-organized and very interesting manuscript. The authors have analyzed three categories of parasitology, i.e. pathological, morphological and molecular level investigations, for Didymozoidae in orange-spotted grouper hosts, and generated a comprehensive report. However, there are few questions need to be answered. If the authors can adequately answer these questions, I will recommend accept the manuscript for publication.

1. Regarding the histopathological analyses (lines 133-137), when you classified tissues as infected, did you examine the entire nasal cavity, or just specific portions of the nasal chamber? How did you determine which infected tissues for sampling, particularly in terms of their size and shape? What was your protocol for processing these tissue samples for the subsequent histological examinations?

2. For the molecular identification of the parasites, I could not find any information about your biological and technical replicates. To obtain a statistically valid picture of genetic variation, you need to balance breadth (sampling different hosts from different locations) with depth (analyzing multiple parasites from each host). So, how many individual worms did you collect from each host? And, how many hosts in total were sampled for rDNA sequencing? Did you pool samples from multiple worms together, or was each worm sequenced individually? Also, how many sequencing replicates did you perform for each sample to ensure reproducibility?

3. The phylogenetic analysis relies exclusively on the 28S rDNA D1 - D3 regions. However, using solely this molecular marker may be problematic:

1. Many didymozoids have undergone recent divergence as they adapted to specific microhabitats within particular fish hosts, which result in cryptic species complexes (rDNA sequences suggest organisms are the same, while their biological characteristics indicate they are different).

2. If these parasites evolved rapidly, it creates rapid radiation. In such cases, the branches representing different species in a 28S rDNA phylogenetic tree become extremely short, making it difficult to resolve relationships between closely related species.

3. Some didymozoids possess unusually long expansion segments in their 28S rDNA, (especially D2 and D3). In such lineages, these expansion segments can evolve at very slow substitution rates, resulting in highly conserved LSU sequences across multiple species. This conservation means different species may share identical or nearly identical D1 - D3 sequences, limiting resolution to the genus level rather than enabling species level discrimination. Moreover, when LSU sequences are extremely conserved, D1 - D3 sequencing lacks sufficient variation to resolve fine scale evolutionary divergences, such as population level genetic structure.

Given these limitations, what is your rationale for concluding that 28S D1 - D3 rDNA sequencing alone provides sufficient resolution to construct a comprehensive phylogenetic framework for Didymozoidae? Will you consider a multi marker approach, such as ITS2 (rDNA spacer, which help resolving species/sibling species relationships) and COI (mtDNA, which help detecting shallow divergences and identify cryptic populations), to create a concatenated phylogenetic tree would provide a more complete picture of Didymozoidae evolutionary history?

These comments are also included in my attached reviewer report.

6. PLOS authors have the option to publish the peer review history of their article (what does this mean?). If published, this will include your full peer review and any attached files.). If published, this will include your full peer review and any attached files.

.

Reviewer #1: No

Reviewer #2: No

To ensure your figures meet our technical requirements, please review our figure guidelines: s://journals.plos.org/plosone/s/figures

You may also use PLOS’s free figure tool, NAAS, to help you prepare publication quality figures: s://journals.plos.org/plosone/s/figures#loc-tools-for-figure-preparation.

---

## [Author Response · Author response to Decision Letter 1]

30 Jan 2026

Response to Reviewers' Comments

Article: PONE-D-25-51753

Original Title: Pathological, morphological, and molecular analysis of didymozoid trematodes infection in the nasal cavity of orange-spotted grouper (Epinephelus coioides) caught from Arabian Gulf waters.

Revised title: "Morphological, molecular, and pathological characterization of didymozoid trematode infection in the nasal cavity of orange-spotted grouper (Epinephelus coioides) from Arabian Gulf waters"

We sincerely thank both reviewers for their constructive feedback and insightful comments, which have significantly improved the quality of our manuscript. Below, we provide detailed point-by-point responses to all comments raised.

Reviewer 1:

We thank Reviewer 1 for the thorough evaluation of our manuscript and for providing valuable suggestions that have helped us improve the clarity and completeness of our work. We have addressed all comments as detailed below.

Comment 1:

Line 128 and 133: Why were the didymozoid cysts fixed in 5% neutral buffered formol saline instead of 10% neutral buffered formalin, as used for the infected tissues?

Response:

We appreciate the reviewer's attention to this methodological detail. The didymozoid cysts were fixed in 5% neutral buffered formol saline to preserve the delicate cyst capsule structure without over-hardening for 24 hours, for parasitological identification. In contrast, the infected tissues for histopathological examination were fixed in standard 10% neutral buffered formalin to ensure optimal tissue preservation and fixation for routine histological processing and staining. We have clarified this methodological distinction in the revised Materials and Methods section.

Comment 2:

Line 165: Please specify the expected and/or obtained PCR product size.

Response:

Thank you for this suggestion. The expected PCR product size for the 28S rDNA D1-D3 region amplification was approximately 1,000-1,100 bp. Our obtained PCR products were 1,042 bp, which is consistent with this expected range and confirms successful amplification of the target region. This information has been added to the Materials and Methods section.

Comment 3:

Line 196-197: Since no external signs of inflammation or tissue damage were observed around the cyst attachment sites, does the relationship between the didymozoid trematodes and the orange-spotted groupers represent a parasitic, mutualistic, or commensal association?

Response:

This is an excellent question that raises important considerations about host-parasite relationships. Actually, didymozoid trematodes are obligatory parasites infecting fish; sometimes, it encapsulated externally on nasal cavity (as recorded in our study) or inside the lamellar tissue of the gills, or even in fish musculature, as recoded by many other studies. In all cases, the constituents of the capsule (mainly connective tissue elements) are derived from the host tissue, with variable harmful effects on the sites of their attachment, depending on the severity of tissue reaction and pathogenicity of the parasite. This host-parasite interaction (parasitism) is differing from mutualistic (a relationship where both organisms benefit from living together), and commensal relationships (a relationship where one organism benefits, while the other is neither significantly helped nor harmed). Fish parasites, particularly certain helminth larvae, can be encapsulated by fibrous capsules without inducing a strong, immediate inflammatory reaction. This mechanism often allows the parasite to remain viable for long periods. While some encapsulation acts as a defence to limit tissue damage, it often represents a passive co-existence.

Comment 4:

Line 2: Since no external signs of inflammation or tissue damage were observed, the authors may need to reconsider the use of the terms "pathological" and "infection" in the title.

Response:

We appreciate the reviewer's concern about terminology accuracy. We have changed the title as following:"Morphological, molecular, and histopathological characterization of didymozoid trematode infection in the nasal cavity of orange-spotted grouper (Epinephelus coioides) from Arabian Gulf waters".

We have replaced the broader term "pathological" with the more precise term "histopathological," which accurately reflects our systematic microscopic examination of tissue-level changes. Regarding the term "infection," we respectfully maintain its use as it is appropriate from a parasitological perspective. Didymozoids are obligate parasites that invade and establish residence in host tissues, constituting a true parasitic infection by standard parasitological definition, regardless of pathology severity.

Comment 5:

Is there any data available on the physiological parameters (weights and lengths) of orange-spotted groupers with and without didymozoid cysts? Such information could help determine whether the presence of didymozoid cysts affects the growth and health of orange-spotted groupers.

Response:

This is an excellent suggestion that would have provided valuable insights into the impact of didymozoid infection on host fitness. Unfortunately, we did not systematically collect morphometric data (weight and length measurements) comparing infected versus uninfected fish during this study. We fully agree that such comparative analyses of condition factors, growth parameters, and overall health indices would be valuable for assessing the biological significance of these parasitic infections. We have incorporated this recommendation into our future research directions in the revised manuscript.

Comment 6:

Line 58-66: Please provide more information about the family Didymozoidae and didymozoid trematodes, including aspects such as their pathogenesis, impact on the host, and associated mortality in host.

Response:

We thank the reviewer for this important suggestion. We have substantially expanded the Introduction section to provide comprehensive background information on Didymozoidae biology and pathological impacts (lines 96-110). These additions provide readers with essential context for understanding the biological and economic importance of didymozoid in Arabian Gulf waters.

Reviewer 2

We are grateful to Reviewer 2 for the detailed and constructive evaluation of our manuscript. The reviewer's expertise in molecular parasitology and integrative taxonomy has been invaluable in strengthening our work. We have carefully addressed all comments and believe the manuscript is substantially improved as a result.

Comment 1:

Regarding the histopathological analyses (lines 133-137), when you classified tissues as infected, did you examine the entire nasal cavity, or just specific portions of the nasal chamber? How did you determine which infected tissues for sampling, particularly in terms of their size and shape? What was your protocol for processing these tissue samples for the subsequent histological examinations?

Response:

We appreciate the reviewer's request for clarification on our histopathological sampling methodology; the existed cysts containing the trematodes were detached carefully and fixed in the proper fixative for further tissue processing. We agree with you in that, the site of attachment on the nasal surface were very difficult to recognize grossly for tissue sampling. So, the detection of these sites of parasites’ attachment was easily recognized on the surface of the cysts, where remnants of the nasal tissue were detached and adhered on the cyst. In this regard, the characteristic epithelial cells of the nasal tissue were clearly demonstrated in Fig. 2B (See letters, ht). So, further future studies are required on the life cycle and pathogenesis of this parasite to determine the specific sites of their attachment on nasal mucosa, factors influencing this attachment process and microscopic tissue alterations in whole nasal tissue of the host.

Comment 2:

For the molecular identification of the parasites, I could not find any information about your biological and technical replicates. To obtain a statistically valid picture of genetic variation, you need to balance breadth (sampling different hosts from different locations) with depth (analyzing multiple parasites from each host). So, how many individual worms did you collect from each host? And, how many hosts in total were sampled for rDNA sequencing? Did you pool samples from multiple worms together, or was each worm sequenced individually? Also, how many sequencing replicates did you perform for each sample to ensure reproducibility?

Response:

We thank the reviewer for highlighting the importance of clearly documenting our sampling strategy and replication approach. We analyzed four individual didymozoid specimens from four different E. coioides hosts collected across different seasons—each processed separately, not pooled. PCR was performed in duplicate for each specimen, and all amplicons were bidirectionally sequenced. Sequence consistency between replicates was confirmed before analysis. All four specimens yielded high-quality sequences (1,042 bp, GenBank: PQ736510-PQ736513), revealing two haplotypes (99.81% identity). Our seasonal sampling strategy prioritized breadth to establish baseline molecular identification for this previously uncharacterized system and detect temporal variation. We acknowledge that deeper within-host sampling would strengthen population genetics analyses and have added this to our limitations section.

Comment 3:

The phylogenetic analysis relies exclusively on the 28S rDNA D1-D3 regions. However, using solely this molecular marker may be problematic: (1) Many didymozoids have undergone recent divergence as they adapted to specific microhabitats within particular fish hosts, which result in cryptic species complexes (rDNA sequences suggest organisms are the same, while their biological characteristics indicate they are different). (2) If these parasites evolved rapidly, it creates rapid radiation. In such cases, the branches representing different species in a 28S rDNA phylogenetic tree become extremely short, making it difficult to resolve relationships between closely related species. (3) Some didymozoids possess unusually long expansion segments in their 28S rDNA (especially D2 and D3). In such lineages, these expansion segments can evolve at very slow substitution rates, resulting in highly conserved LSU sequences across multiple species. This conservation means different species may share identical or nearly identical D1-D3 sequences, limiting resolution to the genus level rather than enabling species-level discrimination. Moreover, when LSU sequences are extremely conserved, D1-D3 sequencing lacks sufficient variation to resolve fine-scale evolutionary divergences, such as population-level genetic structure. Given these limitations, what is your rationale for concluding that 28S D1-D3 rDNA sequencing alone provides sufficient resolution to construct a comprehensive phylogenetic framework for Didymozoidae? Will you consider a multi-marker approach, such as ITS2 (rDNA spacer, which help resolving species/sibling species relationships) and COI (mtDNA, which help detecting shallow divergences and identify cryptic populations), to create a concatenated phylogenetic tree would provide a more complete picture of Didymozoidae evolutionary history?

Response:

We acknowledge these important limitations of single-marker phylogenetics. The 28S rDNA marker was selected based on established use in didymozoid systematics and available reference sequences for comparative analysis. While this marker successfully placed our specimens within Didymodiclinus (99.52% similarity to congeneric species) and revealed intraspecific variation (two haplotypes, 99.81% identity), we agree it has known limitations for resolving closely related species and detecting cryptic diversity. We have revised our discussion to explicitly address these constraints and clarify that our analysis provides robust genus-level identification and phylogenetic placement rather than definitive species delimitation. We now recommend multi-marker approaches (ITS2, COI) as essential future work for fine-scale population structure assessment and cryptic species detection. This recommendation has been added to both the discussion and study limitations sections, emphasizing that integrating multiple loci would provide the resolution necessary to address the specific challenges you outline regarding rapid radiation, conserved expansion segments, and cryptic species complexes in Didymozoidae.

---

## [Decision Letter · Decision Letter 1]

9 Feb 2026

Morphological, molecular, and pathological characterization of didymozoid trematode infection in the nasal cavity of orange-spotted grouper (Epinephelus coioides) from Arabian Gulf waters

PONE-D-25-51753R1

Dear Dr. Attia,

We’re pleased to inform you that your manuscript has been judged scientifically suitable for publication and will be formally accepted for publication once it meets all outstanding technical requirements.

An invoice will be generated when your article is formally accepted. Please note, if your institution has a publishing partnership with PLOS and your article meets the relevant criteria, all or part of your publication costs will be covered. Please make sure your user information is up-to-date by logging into Editorial Manager at Editorial Manager® and clicking the ‘Update My Information' link at the top of the page. For questions related to billing, please contact  and clicking the ‘Update My Information' link at the top of the page. For questions related to billing, please contact billing support..

Kind regards,

Tzong-Yueh Chen, Ph.D.

Academic Editor

PLOS One

Additional Editor Comments (optional):

Reviewers' comments:

Reviewer's Responses to Questions

**Comments to the Author**

1. If the authors have adequately addressed your comments raised in a previous round of review and you feel that this manuscript is now acceptable for publication, you may indicate that here to bypass the “Comments to the Author” section, enter your conflict of interest statement in the “Confidential to Editor” section, and submit your "Accept" recommendation.

Reviewer #1: All comments have been addressed

Reviewer #2: All comments have been addressed

2. Is the manuscript technically sound, and do the data support the conclusions?

Reviewer #1: Yes

Reviewer #2: Yes

3. Has the statistical analysis been performed appropriately and rigorously? 

Reviewer #1: Yes

Reviewer #2: Yes

4. Have the authors made all data underlying the findings in their manuscript fully available?

Reviewer #1: Yes

Reviewer #2: Yes

5. Is the manuscript presented in an intelligible fashion and written in standard English?

Reviewer #1: Yes

Reviewer #2: Yes

6. Review Comments to the Author

Reviewer #1: 1. All comments have been addressed.

2. Line 160: The description "fixed in . Tissue blocks" should be deleted.

Reviewer #2: Since the authors has adequately answered my questions and provided reasonable rationalizations; and it is a first in class study in certain field, I will recommend accept the manuscript for publication.

7. PLOS authors have the option to publish the peer review history of their article (what does this mean?). If published, this will include your full peer review and any attached files.). If published, this will include your full peer review and any attached files.

.

Reviewer #1: No

Reviewer #2: No

---

## [Editor Report · Acceptance letter]

PONE-D-25-51753R1

PLOS One

Dear Dr. Attia,

I'm pleased to inform you that your manuscript has been deemed suitable for publication in PLOS One. Congratulations! Your manuscript is now being handed over to our production team.

Kind regards,

on behalf of

Prof. Tzong-Yueh Chen

Academic Editor

PLOS One